# Photo-driven redox-neutral decarboxylative carbon-hydrogen trifluoromethylation of (hetero)arenes with trifluoroacetic acid

Jin Lin[1], Zhi Li[2], Jian Kan[1], Shijun Huang[1], Weiping Su[1] & Yadong Li[2]

Catalytic oxidative C–H bond functionalization reactions that proceed without requiring stoichiometric amounts of external oxidants or pre-functionalized oxidizing reagents could maximize the atom- and step-economy in chemical syntheses. However, such a transformation remains elusive. Here, we report that a photo-driven catalytic process enables decarboxylative C–H trifluoromethylation of (hetero)arenes with trifluoroacetic acid as a trifluoromethyl source in good yields in the presence of an external oxidant in far lower than stoichiometric amounts (for example, 0.2 equivalents of $Na_2S_2O_8$) using Rh-modified $TiO_2$ nanoparticles as a photocatalyst, in which $H_2$ release is an important driving force for the reaction. Our findings not only provide an approach to accessing valuable decarboxylative C–H trifluoromethylations via activation of abundant but inert trifluoroacetic acid towards oxidative decarboxylation and trifluoromethyl radical formation, but also demonstrate that a photo-driven catalytic process is a promising way to achieve external oxidant-free C–H functionalization reactions.

[1] State Key Laboratory of Structural Chemistry, Fujian Institute of Research on the Structure of Matter, Chinese Academy of Sciences, Fujian 350002, China. [2] Department of Chemistry, Tsinghua University, Beijing 100084, China. Correspondence and requests for materials should be addressed to W.S. (email: wpsu@fjirsm.ac.cn) or to Y.L. (email: ydli@tsinghua.edu.cn).

C–H bond functionalization, due to its simplified synthetic routes and readily available starting materials, provides a potentially atom- and step-economic synthetic approach complementary to the conventional functionality-based synthetic methods[1–5]. The potential power of the C–H functionalization strategy in organic synthesis has been clearly demonstrated by its application to natural product synthesis[6], medicine synthesis[7,8], late-stage functionalization of complex molecules[9,10] and material preparations[11]. Full realization of the intrinsic advantage of the atom- and step-economy of C–H functionalization reactions, however, remains a significant challenge because such reactions generally require either stoichiometric amounts of external oxidants (Fig. 1a) such as metal salt oxidants or oxidizing reactants that are often prepared via multistep procedures (Fig. 1b). Natural photosynthesis, a process that enables photo-driven water splitting to molecular dioxygen and atomic hydrogen via coupling of the water oxidation half-reaction with the proton reduction half-reaction[12], offers a blueprint for the design of oxidative C–H functionalization reactions that proceed without reliance on any external oxidant or oxidizing reactant.

On the other hand, the importance of trifluoromethyl ($CF_3$) group-containing compounds in pharmaceuticals, agrochemicals and materials has spurred tremendous efforts devoted to the development of methods for the incorporation of trifluoromethyl group into molecular frameworks and the discoveries of trifluoromethylating reagents[13]. As a result of the inventions of a diverse range of trifluoromethylating reagents including nucleophilic ones ($CF_3^-$) and electrophilic ones ($CF_3^+$), a variety of metal-catalysed direct C–H trifluoromethylations have been achieved[14–18]. Benefiting from the identification of various precursors that are capable of generating trifluoromethyl radicals through regular oxidation[19] or photoredox catalysis[20] under mild conditions, the radical-based reactions have evolved into an efficient approach to construct C-$CF_3$ bonds. In spite of these remarkable advances, the commonly used trifluoromethylating reagents are very expensive due to their time-consuming and multistep syntheses, which prompt chemists to develop readily available, cost-effective surrogates of existing trifluoromethylating reagents.

Herein, we report a photo-driven decarboxylative C–H trifluoromethylation of nonactivated (hetero)arenes with readily available, low-cost trifluoroacetic acid as a trifluoromethylating reagent. Using Rh-modified $TiO_2$ NPs as a catalyst, our reaction can occur in the absence of external oxidant to afford a trifluoromethylation product in a modest yield with $H_2$ release as a driving force for the reaction. Our findings demonstrate that the underlying redox chemistry of natural photosynthesis can operate in C–H functionalization reactions, which may provide an opportunity to maximize the step- and atom-economy of such reactions.

## Results

**Reaction design**. Minisci reactions enable the use of alkyl carboxylic acids for C–H alkylation of electron-deficient pyridines via oxidative decarboxylation of alkyl carboxylic acids[21]. Recently, Minisci reaction of aryl carboxylic acids has been realized by us[22]. Minisci reactions have implied that trifluoroacetic acid (TFA) may act as a trifluoromethylating reagent via oxidative decarboxylation to generate trifluoromethyl radical[23-25]. In terms of its stability, ready availability and low price, trifluoroacetic acid is an intrinsically advantageous trifluoromethylating reagent. The extraordinary inertness of trifluoroacetic acid (electro-oxidation of TFA anion to generate trifluoromethyl radical occurs at potentials higher than $+2.24\,V$ versus the saturated calomel electrode (SCE)[26]), however, largely retards its application to the trifloromethylation reactions. For example, the decarboxylative C–H trifluoromethylation mediated by a stoichiometric amount silver salt required 2 equivalents of $K_2S_2O_8$ as oxidant and tolerated only a few electron-deficient substituted benzenes[25].

A recent report on using trifluoroacetic anhydride, a derivative of trifluoroacetic acid, for the C–H trifluoromethylation provides an alternative solution for the inexpensive trifluoromethyl source[27], which, however, requires the use of stoichiometric pyridine N-oxide as a sacrificial oxidant.

$TiO_2$-based NPs are extensively used as photocatalysts for the photo-driven water splitting processes[28]. Significant redox potential ($+3.1\,V$ versus SCE)[29] of the photo-generated hole in

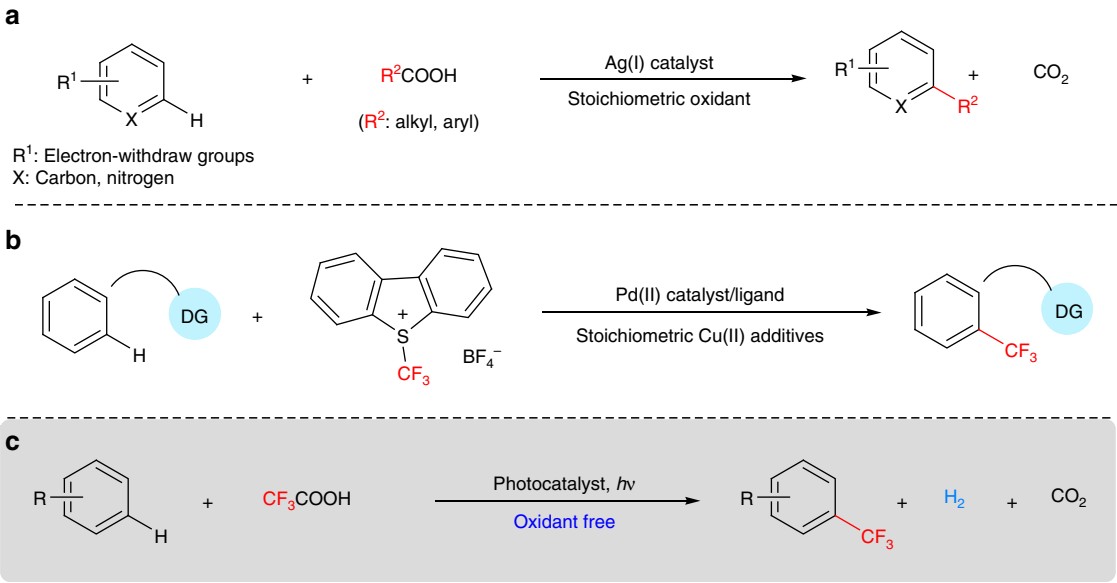

**Figure 1 | Catalytic oxidative C–H functionalization.** (**a**) Minisci reaction: the example of the C–H functionalization with stoichiometric external oxidant. (**b**) C–H trifluoromethylation with Umemoto reagent: the example of C–H functionalization with pre-synthesized oxidazing reagent. (**c**) Photo-driven C–H trifluoromethylation without the reliance on external oxidant (this work).

**Table 1 | Effects of reaction parameters on photo-driven decarboxylative C–H trifluoromethylation.**

Trifluoromethylation

365 nm irradiation, TFA, room temperature, 24 h, commercial anatase 60 nm $TiO_2$ (20 mol%)

Initial conditions

| Entry | Variation from the initial conditions | Yield |
|---|---|---|
| 1 | Initial conditions | 6 |
| 2 | 0.1 wt% Rh/anatase $TiO_2$ instead of $TiO_2$ | 15 |
| 3 | 0.1 wt% Rh/anatase $TiO_2$ instead of $TiO_2$, $Na_2SO_4$ (2 equiv.) as additive | 51* |
| 4 | 0.1 wt% Rh/anatase $TiO_2$ instead of $TiO_2$, $Na_2S_2O_8$ (0.2 equiv.) as additive | 70† |
| 5 | Visible light instead of 365 nm ultraviolet | Trace |
| 6 | $CF_3COONa$ (1 equiv.) as additive | Trace |
| 7 | No $TiO_2$ | no reaction |
| 8 | No light | no reaction |

Initial conditions: benzene (0.5 mmol), photocatalyst (20 mol% based on the amount of $TiO_2$ relative to benzene), TFA (15 ml), 365 nm ultraviolet irradiation (250 W), room temperature, 24 h. $^{19}F$ NMR yields using 1-methoxy-4-(trifluoromethoxy)benzene as the internal standard.
*$2a$/$2aa$-$1$/$2aa$-$2$/$2aa$-$3$ = 8.3/1/2.4/2.9.
†$2a$/$2aa$-$1$/$2aa$-$2$/$2aa$-$3$ = 17.3/2.8/1/1.8.

the valence band of $TiO_2$, in principle, would make it capable of oxidizing trifluoroacetic acid to generate trifluoromethyl radicals via decarboxylation and therefore trigger the trifluoromethylation reaction. Accordingly, we envisioned that $TiO_2$-based NPs could catalyse the photo-driven decarboxylative C–H trifluoromethylation reaction with trifluoroacetic acid as a trifluoromethyl source (Fig. 1c). More importantly, the similarity to the photocatalytic water splitting process also implies the possibility that this decarboxylative C–H trifluoromethylation reaction proceeds without the need for the stoichiometric amount of external oxidants. Only one example of $TiO_2$-promoted photo-driven decarboxylative C–H trifluoromethylation reaction of benzenes in trifluoroacetic acid that required stoichiometric amount of silver trifluoroacetate as an oxidant and gave poor yields indicates that there is significant room to improve the photocatalyst[23].

**Establishment of photo-driven trifluoromethylation**. We started our investigations by testing the trifluoromethylation of benzene with trifluoroacetic acid as a trifluoromethyl source. Numerous trials were performed to screen the reaction parameters such as different solvents, $TiO_2$-based photocatalysts, light sources and external oxidants (details are provided in Supplementary Table 1). Selected results are presented in Table 1. Most strikingly, in this process C–H trifluoromethylation can happen with no sacrificial oxidant (entries 1 and 2), with very weak oxidant (entry 3), as well as with the dosage of external oxidant far less than the stoichiometric amount (entry 4).

When the reaction was conducted at room temperature under near-ultraviolet light irradiation from 365 nm Hg lamp (250 W), with 20 mol% commercial available anatase $TiO_2$ NPs with a diameter of 60 nm as the photocatalyst, 6% yield was observed in the absence of external oxidant (entry 1). This reaction was accompanied by $H_2$ gas formation, indicating that coupling of the photo-generated electrons with the protons released from decarboxylative C–H trifluoromethylation reaction occurred. The catalytic efficiency of the anatase $TiO_2$ NPs could be enhanced by loading rhodium metal NPs of 0.1 wt % (relative to the amount of $TiO_2$) onto their surface, which gave rise to 15% yield of trifluoromethylated benzene (entry 2). Interestingly, by employing 2 equivalences of $Na_2SO_4$ as additives, the reaction

occurred smoothly to give the desired product in 51% yield (entry 3). Introduction of substoichiometric external oxidants, such as 0.2 equivalents of $Na_2S_2O_8$ could increase the reaction yield to 70% (entry 4). The attempt to use visible light in place of ultraviolet light failed to give the satisfied result because of the wide band gap of $TiO_2$ (entry 5). We also found that introducing one equivalence of $CF_3COONa$ into the reaction system almost shut down this reaction (entry 6), which presumably arose from the difference in substrate-$TiO_2$ surface interaction between trifluoroacetic acid and $CF_3COONa$. Moreover, control experiments revealed that both anatase $TiO_2$ photocatalyst and ultraviolet light are necessary for the trifluoromethylation reaction to occur (entries 7 and 8).

**Substrate scope and gram-scale reations**. With the established reaction conditions for the photo-driven decarboxylative C–H trifluoromethylation, we evaluated substrate scope of substituted benzenes in the presence of substoichiometric amount of oxidant (0.1–0.4 equivalences of $Na_2S_2O_8$), using trifluoroacetic acid as a trifluoromethylating reagent (for more information, see Supplementary Methods and Supplementary Fig. 53). As shown in Fig. 2, this developed C–H trifluoromethylation reaction was compatible with a broad range of functional groups, such as halogen (2b-2e), keto (2f, 2g), ester (2h, 2i) and nitrile (2j) groups. Interestingly, ultraviolet light-sensitive C-I bond remained intact during the course of C–H trifluoromethylation (2e). And benzoic acid, which is prone to oxidative decarboxylation under oxidative conditions, could undergo trifluoromethylation reaction, albeit in a diminished yield (2l). Disubstituted benzenes (2m, 2n and 2o) could also be trifluoromethylated in synthetically useful yields. Moreover, a variety of nitrogen-containing heteroarenes, such as pyridines (2p, 2q), pyrimidine (2r), pyrazolinopyrimidine (2w) and biologically active xanthines (2s-2v) were amenable to this trifluoromethylation reaction, affording a satisfying outcome. Importantly, the reactions of these nitrogen-containing heteroarenes exhibited a high level of position selectivity, and could be scaled up for the gram-scale synthesis as exemplified by the reactions of caffeine and theophylline (2s, 2t), highlighting the practicality of this approach. In the established reaction, the use of 15 ml trifluoroacetic acid as a solvent may result from the need

**Figure 2 | Photo-driven C–H trifluoromethylation of substituted benzenes and nitrogen-containing heteroarenes.** Standard reaction conditions: substrates (0.5 mmol), 0.1 wt% Rh/anatase $TiO_2$ NPs (20 mol%), $Na_2S_2O_8$ (10–40 mol%), TFA (10–15 ml), 365 nm ultraviolet, room temperature, 24 h. Yields were determined by $^{19}F$ NMR spectrum. †Isolated yield.

for sufficient light-absorbing surface to obtain satisfying yields presumably due to relative low concentration of catalytic active centres of heterogeneous $TiO_2$-based nanocatalyst. Despite this, excess TFA in this reaction is reusable by means of easy distillation for its purification and collection.

The established reaction conditions for arene trifluoromethylation were applicable to the reaction of iodine with trifluoroacetic acid to afford trifluoromethyl iodide ($CF_3I$), a commonly used trifluoromethylating reagent[13], in a 70% yield (Fig. 3a and

Supplementary Fig. 1), which provides an operationally simple approach to the efficient synthesis of $CF_3I$.

**Mechanistic insights and nanocatalyst characterization.** Although the position selectivity of this decarboxylative trifluoromethylation reaction suggested that the reaction would proceed via trifluoromethyl radical formation, further experiments were still required to gain insights into reaction

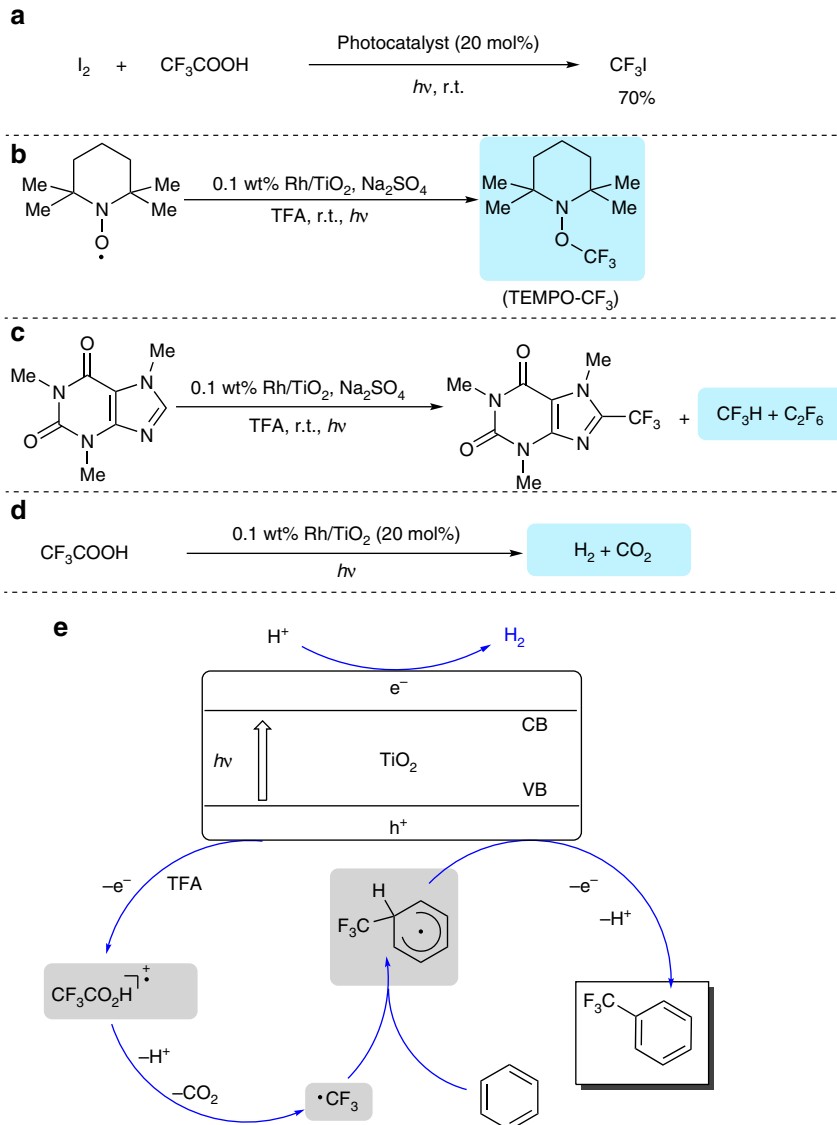

**Figure 3 | Mechanism studies.** (**a**) Synthesis of $CF_3I$ via decarboxylative coupling of $I_2$ with TFA. (**b**) Capture of $CF_3$ radical by TEMPO. (**c**) Determination of $CF_3H$ and $C_2F_6$ side products in trifluoromethylation of caffeine. (**d**) Determination of $H_2$ and $CO_2$ from the photo-driven decarboxylation of TFA. (**e**) The proposed reaction mechanism.

mechanism. As shown in Fig. 3b, on adding 2,2,6,6-tetramethyl-1-piperidinyloxy (TEMPO), a well-known radical scavenger, to trifluoroacetic acid solution under standard photoredox conditions, the trifluoromethyl radicals generated from decarboxylation of trifluoroacetic acid was trapped with TEMPO to form TEMPO-$CF_3$ adduct that was confirmed by $^{19}F$ NMR spectroscopy (Supplementary Fig. 2).

In addition, both trifluoromethane ($CF_3H$) and hexafluoroethane ($C_2F_6$) were also observed as side-products during the trifluoromethylation reaction of caffeine (Fig. 3c and Supplementary Figs 3 and 4), providing further evidence to support that trifluoromethyl radical is the intermediate of the decarboxylative trifluoromethylation reaction. Furthermore, formations of both $H_2$ and $CO_2$ were observed when trifluoroacetic acid was subjected to the standard reaction conditions in the absence of any arene reactant (Fig. 3d and Supplementary Figs 5 and 6), which is in accord with the aforementioned observation in the decarboxylative trifluoromethylation of benzene. The fact that the oxidative decarboxylation of trifluoroacetic acid is concurrent with $H_2$ formation strongly suggested that the redox chemistry of

the photo-driven water splitting to $H_2$ and $O_2$ operates in the decarboxylative C–H trifluoromethylation reaction with trifluoacetic acid as a trifluoromethyl source.

On the basis of mechanistic investigation, a possible reaction pathway was proposed for the $TiO_2$-catalysed photo-driven decarboxylative trifluoromethylation with trifluoroacetic acid as trifluoromethyl source (Fig. 3e). Initially, the oxidation of trifluoroacetic acid by the photo-generated hole in the valence band of $TiO_2$ leads to rapid decarboxylation to form trifluoromethyl radical with carbon dioxide and proton released. The resultant trifluoromethyl radical is then intercepted by benzene to produce trifluoromethyl benzene radial. Finally, this trifluoromethyl benzene radical undergoes oxidation through a single-electron transfer process and subsequent deprotonation to afford a trifluoromethylated benzene. Meanwhile, the released protons accept the photo-generated electron to release hydrogen gas, a process similar to the photo-driven water splitting to $H_2$ evolution[30]. Alternatively, the external oxidant introduced into the reaction system can also accept the photo-generated electron to drive the reaction. Consequent, combination of contributions

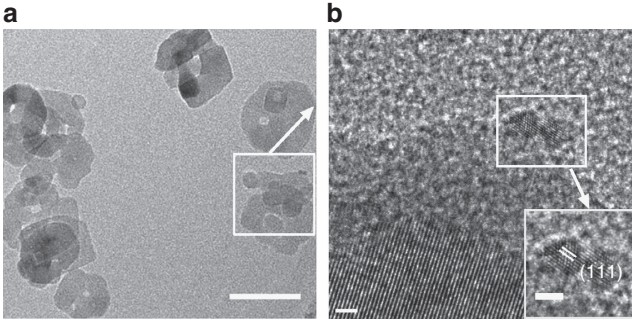

**Figure 4 | Characterization of the nanocatalyst.** (**a**) TEM image of Rh/TiO$_2$ (scale bar, 50 nm), (**b**) HRTEM image (scale bar, 5 nm) and lattice fringe image (inset, scale bar of 2 nm) of Rh(0) NP. HRTEM, high-resolution TEM.

of proton and a substoichiometric amount of external oxidant provided good yields for the photo-driven decarboxylative trifluoromethylation reaction.

The photocatalyst used in this reaction, that is, the Rh-modified TiO$_2$ NPs, were structurally characterized. The inductively coupled plasma mass spectrometry revealed that this Rh/TiO$_2$ photocatalyst contained Rh content of 0.1 wt%. The powder X-ray diffraction pattern of this nanocatalyst could be exclusively indexed to anatase-phase TiO$_2$ (Supplementary Fig. 7). The transmission electron microscopy (TEM) image (Fig. 4a) and high-resolution TEM image (Fig. 4b) showed that the metallic Rh NPs with an average size of 2.04 nm (Supplementary Figs 8–10) was located on the surface of anatase NPs of about 60 nm. Besides, X-ray photoelectron spectroscopy indicated the signals of Rh 3d$_{5/2}$ and 3d$_{3/2}$ with binding energies of 306.42 and 311.27 eV, typical of Rh (0), further verifying that Rh existed in the form of metallic Rh (Supplementary Figs 11–15)[31].

## Discussion

On the basis of the characterization of the Rh-modified anatase TiO$_2$ NPs, we attempt to rationalize the photocatalysis efficiency enhancement associated with the attachment of metallic Rh NPs onto the surface of anatase TiO$_2$ NPs. Presumably, in the Rh-modified anatase NPs, the difference in Fermi levels between metal Rh and semiconductor TiO$_2$ may give rise to the migration of photo-generated electron from TiO$_2$ with higher Fermi level (the work function of TiO$_2$ is 4.8 eV) to Rh with lower Fermi level (the work function of Rh is 5.6 eV) until equilibrium alignment of two Femi levels[32]. As a result, the Schottky barrier is created at the interface between Rh and TiO$_2$ NPs[33]. Such a barrier may act as an effective trap for electron capture, and prevent from recombination between photo-generated electron/hole pairs, which eventually leads to the improvement of catalytic efficiency (shown in Supplementary Fig. 16).

In this work, we have developed a photo-driven C–H trifluoromethylation reaction that enables the use of readily available and cheap trifluoroacetic acid as a trifluoromethyl source for high-valuable trifluoromethylation reactions. Importantly, such a photo-driven reaction could proceed with the dosage of external oxidant far less than the stoichiometric amount or even without any external oxidant, in which H$_2$ release provided an essential driven force for reaction to occur. This reaction demonstrates that the principle underlying the photo-driven water splitting to H$_2$ and O$_2$ is applicable to C–H functionalization reaction, which will inspire the development of photocatalysts to improve efficiencies of both utility of light and coupling of the proton from C–H activation process with the photo-generated electron and ultimately achieve general

approach to external oxidant-free C–H functionalization reactions[34].

## Methods

**Preparation of 0.1 wt% Rh-modified anatase TiO$_2$ nanopaticles.** In a 30 ml quartz round bottom flask, a mixture of 60 nm anatase TiO$_2$ (Aladdin reagent, 3.0 g) and RhCl$_3$•xH$_2$O (0.023 g, Rh content of 39%) was deoxygenated and filled with N$_2$, followed by addition of deoxygenated methanol (30 ml). The suspension was stirred with an irradiation of 250 W high-pressure Hg lamp (365 nm ultra-violet) for 48 h. After illumination, nanocatalysts were separated from the solution by centrifugation (8,000 r.p.m., 2 min, 298 K). The separated nanocatalysts were washed for three times by ethanol and three times by water, dried in vacuum at 298 K for 8 h. The Rh content of as-prepared nanocatalyst was 0.1 wt% by inductively coupled plasma analysis.

**General procedure for photo-driven C–H trifluoromethylation.** The photo-driven C–H trifluoromethylation reaction of caffeine is used as an example. Photocatalyst (0.016 g, 0.2 mmol), caffeine (0.0971 g, 0.5 mmol) and Na$_2$S$_2$O$_8$ (0.0238 g, 0.1 mmol) were introduced into a Schlenk tube. Then, the tube was fitted with a rubber septum. After evacuation and N$_2$ backfill for three times, distilled trifluoroacetic acid (15 ml) was added to the Schlenk tube through the rubber septum using syringes, and the rubber septum was replaced by a Teflon cap under N$_2$ flow. The reaction was performed under illumination of 250 W high-pressure Hg lamp (365 nm ultraviolet) at room temperature for 48 h. After reaction, the trifluoroacetic acid was removed under reduced pressure and the residue was purified by flash chromatography on silica gel (eluent: EtOAc/hexanes) to provide the corresponding product. The yield were also determined by [19]F NMR spectrum using 1-methoxy-4-(trifluoromethoxy)benzene (76 µl, 0.5 mmol, δ–58.4 p.p.m.) as an internal reference. For NMR spectra of the compounds in this article, see Supplementary Figs 17–52.

**Data availability.** The data supporting this study are available from the corresponding author upon reasonable request.

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

## Acknowledgements

Financial supports from the NSFC (21431008, 21332001, 21403240, 21301173, 21401197 and u1505242), the CAS/SAFEA International Partnership Program for Creative Research Teams and the Strategic Priority Research Program of the Chinese Academy of Sciences (XDB20000000) are greatly appreciated. We thank Prof Xinchen Wang at Fuzhou University for $H_2$ determination assisstance, and Prof Lei Liu and Prof Wei He at Tsinghua University for participating in discussion of this project.

## Author contributions

J.L., Z.L., J.K. and S.H. performed and analysed experiments. W.S. and Y.L. designed the experiments and also prepared this manuscript.

## Additional information

**Competing financial interests:** The authors declare no competing financial interests.

