## [Peer Review File · Nature Communications]

Reviewers' comments:

Reviewer #1 (Remarks to the Author):

This manuscript concerns the synthesis of trifluoromethylated derivatives via decarboxylative C-H using Rh-modified TiO₂ as photo-catalyst and Na₂S₂O₈ as an external oxidant in less than stoichiometric amount. The manuscript is a result of an ongoing work of authors and others on the synthesis and performance of photo-catalysts and redox reactions. This work, therefore, limits the novelty aspects a bit although interesting results have been described on the photo-catalytic performance of the Rh-modified TiO₂. The descriptions on the experimental data are well organized as a communication. In conclusion, this manuscript could be acceptable for the publication in Nature Communication after appropriate revision.

1. Kindly check the manuscript for all typographical errors. Authors may like to remove certain repetitive sentences and be little less verbose; they can start under the heading, Results.
2. What is the role of Rh in reaction mechanism? For better understanding, the author should delineate the role of Rh while discussing mechanism of the reaction; the role of TiO₂ in reaction mechanism is pretty obvious.
3. Please improve the quality of figures from Fig. 3 (A) to 3 (E). All figures appear to be blurry.
4. This reviewer would suggest the authors to take into consideration in the "Introduction Section" the following references which deals with the C-H activation and trifluoromethylation (Org. Lett., 2016, 18, 2906–2909; ACS Sustainable Chemistry & Engineering 2016, 4, 2333-2336; Org. Lett., 2015, 17, 4702–4705; Green Chemistry 2016, 18 (1), 251-254; J. Am. Chem. Soc., 2012, 134, 9034–9037; Chemical Communications 2015, 51, 15554-15557; ChemCatChem 2011, 3, 1329-1332).

Reviewer #2 (Remarks to the Author):

In this manuscript, Li and Su and co-workers reported a decarboxylative trifluoromethylation protocol by using TFA as the CF₃ source. This reaction features using TiO₂ as the sensitizer, redox neutral and promoted by using UV light at room temperature.

Using TFA (or trifluoroacetate) as the CF₃ radical source is not a new idea. As cited by the authors, reference 22 showed an oxidative trifluoromethylation protocol by using stoichiometric amount of persulfate through TFA as the CF₃ source with Ag as the catalyst. Compared to reference 22, I think this presented work bears advantages such as without using Ag salt and less stoichiometric amount of persulfate. However, the downside of this work is using TiO₂ photosensitizer and UV light as the catalyst and energy input. In this regard, using TiO₂ combined with UV to get the CF₃ radical from TFA and its salt is not new either. As shown in reference 20, the authors used TiO₂ and UV to get the trifluoromethylation product of (Hetero)arenes by using TFAAg salt and TFA as the CF₃ source. Compared to reference 20, this work employed only TFA instead of TFA silver salt, which is definitely one advantage. Instead of Ag, the additional Rh was added probably in favor of the H₂ formation. Releasing H₂ gas instead of adding external oxidant such as persulfate in reference 22 and Ag salt in reference 20 is the most important contribution of this paper, in my opinion. But at least, the authors should discuss and compare their work with both works mentioned above (references 20 and 22) in the manuscript.

From the perspective of a synthetic chemist, this protocol is not user friendly. In order to convert 0.5 mmol substrate, 15 mL TFA (a very nasty and strong acid) has to be used as the solvent. I am wondering if the authors can screen a more regular solvent instead of TFA. Once that solvent is found, I would be happy to recommend the acceptance of this paper in Nature Communication considering the novelty of this work itself

Reviewer #3 (Remarks to the Author):

Su and Li reported a practical and novel photo-driven C-H trifluoromethylation reaction, which applied trifluoroacetic acid as trifluoromethyl source. Rh-modified TiO₂ nanoparticle as catalysis made the transformation occurred in the absence of stoichiometric amounts of external oxidants. It was well demonstrated that a photo-driven catalysis process is a promising way to achieve external oxidant-free C-H functionalization reactions. However, there are still some problems should be answered before the manuscript was accepted on Nat. Commun.

1. Although no stoichiometric amounts of oxidant are necessary for the reaction, excess trifluoroacetic acid (15 mL) is needed. Please state the reason in manuscript (SI, Table S1, entry 2-17).

2. The effects of substoichiometric amounts of oxidant should be explained in the mechanism (Figure 3, E) as a key advantage.

3. The author declared that "the released protons accept the photo-generated electron to release hydrogen gas, a process similar to the photo-driven water splitting to H₂ evolution" (line 225 to 227). Here, the representative references should be included.

Response to reviewers' comments

Response to the comments of reviewer #1

Comment #1. *This manuscript concerns the synthesis of trifluoromethylated derivatives via decarboxylative C-H using Rh-modified TiO₂ as photo-catalyst and Na₂S₂O₈ as an external oxidant in less than stoichiometric amount. The manuscript is a result of an ongoing work of authors and others on the synthesis and performance of photo-catalysts and redox reactions. This work, therefore, limits the novelty aspects a bit although interesting results have been described on the photo-catalytic performance of the Rh-modified TiO₂. The descriptions on the experimental data are well organized as a communication. In conclusion, this manuscript could be acceptable for the publication in Nature Communication after appropriate revision.*

Response: We greatly appreciate the reviewer for the favorable comments on the manuscript. Based on the reviewer's helpful comments and suggestions, we have made all requested changes during revision of this manuscript, which were highlighted with a yellow background in the revised manuscript and supplementary information.

Comment #2. *Kindly check the manuscript for all typographical errors. Authors may like to remove certain repetitive sentences and be little less verbose; they can start under the heading, Results.*

Response: We have done our best to find all typographical errors and correct them, polish English of this manuscript, please see the corresponding revisions in the revised manuscript.

Comment #3. *What is the role of Rh in reaction mechanism? For better understanding, the author should delineate the role of Rh while discussing mechanism of the reaction; the role of TiO₂ in reaction mechanism is pretty obvious.*

Response: The transmission electron microscopy (TEM) image and high-resolution TEM image showed that the metallic Rh nanoparticles with an average size of 2.04 nm was located on the surface of anatase nanoparticles of about 60 nm. the difference in Fermi levels between metal Rh and semiconductor TiO₂ may give rise to the migration of photogenerated electron from TiO₂ with higher Fermi level (the work function of TiO₂ is 4.8 eV) to Rh with

lower Fermi level (the work function of Rh is 5.6 eV) until equilibrium alignment of two Fermi levels (please see reference: *J. Phys. Chem.* 1994, **98**, 7147-7152). As a result, the Schottky barrier is created at the interface between Rh and TiO₂ nanoparticles (please see reference: *J. Phys. Chem. C* 2010, **114**, 17660-17664). Such a barrier may act as an effective trap for electron capture, and prevent from recombination between photogenerated electron/hole pairs, which eventually leads to the improvement of catalytic efficiency

In response to this comment, we have added some sentences to the revised manuscript to explain the role of Rh in nanocatalyst, as well as a Scheme to supplementary material to illustrate principle of the Schottky barrier formation.

Comment #4. *Please improve the quality of figures from Fig. 3 (A) to 3 (E). All figures appear to be blurry.*

Response: Thank you very much for pointing out this problem. We have improving the quality of all figures in the revised manuscript.

Comment #5. *This reviewer would suggest the authors to take into consideration in the "Introduction Section" the following references which deals with the C-H activation and trifluoromethylation (*Org. Lett.*, 2016, 18, 2906–2909; *ACS Sustainable Chemistry & Engineering* 2016, 4, 2333-2336; *Org. Lett.*, 2015, 17, 4702–4705; *Green Chemistry* 2016, 18 (1), 251-254; *J. Am. Chem. Soc.*, 2012, 134, 9034–9037; *Chemical Communications* 2015, 51, 15554-15557; *ChemCatChem* 2011, 3, 1329-1332).*

Response: .We have cited these references suggested by reviewer at “introduction Section” and other appropriate positions.

Response to the comments of reviewer #2

Comment #1. *In this manuscript, Li and Su and co-workers reported a decarboxylative trifluoromethylation protocol by using TFA as the CF₃ source. This reaction features using TiO₂ as the sensitizer, redox neutral and promoted by using UV light at room temperature. Using TFA (or trifluoroacetate) as the CF₃ radical source is not a new idea. As cited by the authors, reference 22 showed an oxidative trifluoromethylation protocol by using stoichiometric amount of persulfate through TFA as the CF₃ source with Ag as the catalyst. Compared to reference 22, I think this presented work bears advantages such as without using Ag salt and less stoichiometric amount of persulfate. However, the downside of this work is using TiO₂ photosensitizer and UV light as the catalyst and energy input. In this regard, using TiO₂ combined with UV to get the CF₃ radical from TFA and its salt is not new either. As shown in reference 20, the authors used TiO₂ and UV to get the trifluoromethylation product of (Hetero)arenes by using TFAAg salt and TFA as the CF₃ source. Compared to reference 20, this work employed only TFA instead of TFA silver salt, which is definitely one advantage. Instead of Ag, the additional Rh was added probably in favor of the H₂ formation. Releasing H₂ gas instead of adding external oxidant such as persulfate in reference 22 and Ag salt in reference 20 is the most important contribution of this paper, in my opinion. But at least, the authors should discuss and compare their work with both works mentioned above (references 20 and 22) in the manuscript.*

Response: We greatly appreciate the reviewer for the favorable comments on the novelty of our work. As agreed by the reviewer, the advantage of our work is that our C-H trifluoromethylation reaction could occur without external oxidant, and produce the desired trifluoromethylated products in good yield in the presence of only substoichiometric amount of external oxidant. In both cases, hydrogen gas formation via coupling of proton with photo-generated electron provided an important driven force for reactions. As a result, our work demonstrates that the photo-driven process has a great potential to enable external oxidant-free C-H functionalization reaction and therefore realize the advantage of atom-economy of direct C-H functionalization.

Indeed, two precedents for the decarboxylative C-H trifluoromethylation using trifluoroacetic acid (TFA), which were cited as references 20 and 22 in our manuscript, have been reported. One is the TiO₂-catalyzed photodriven decarboxylative C-H trifluoromethylation reaction of benzenes that gave poor yields (the highest yield of 50%) using stoichiometric amount of silver trifluoroacetate and trifluoroacetic acid as trifluoromethyl sources. The other is the stoichiometric silver mediated decarboxylative trifluoromethylation reaction of benzene using trifluoroacetic acid as trifluoromethyl source that was limited to electron-deficient benzenes bearing five kinds of substituents and required 2 equivalents of

$\text{K}_2\text{S}_2\text{O}_8$ as oxidant (40 mol% Ag_2CO_3 relative to benzene). In response to this comment, we have added some sentences to discuss these two precedent reactions with a yellow background (please see the revised manuscript).

Our group has always been interested in the decarboxylative C-H functionalization reactions. Particularly, we have recently achieved a general silver-catalyzed method for decarboxylative C-H arylation via oxidative decarboxylation, which prompted us to explore the possibility of the use of trifluoroacetic acid as a trifluoromethyl source for decarboxylative C-H trifluoromethylation in view of the importance of trifluoromethylation reaction and the readily availability of trifluoroacetic acid. Considering that trifluoroacetic acid is very stable towards decarboxylation to form reactive trifluoromethyl radical, we designed photocatalytic process to realize the goal. Although literature survey showed that the photocatalytic decarboxylative trifluoromethylation of benzenes under TiO_2 catalyst had been reported, the poor yields of the reaction indicated that the great room remained for improvement of this reaction. Finally, we improved this reaction by using Rh-modified TiO_2 nanocatalyst. Surprisingly, our photo-driven trifluoromethylation reaction could occur in the presence of substoichiometric amount of external oxidant, or even without external oxidant because protons could accept the photo-generated electrons. As such, our findings may represent a progress in the development of catalytic method for decarboxylative trifluoromethylation reaction of trifluoroacetic acid, and may have an impact on the design of external oxidant-free C-H functionalization reaction.

Comment #2. *From the perspective of a synthetic chemist, this protocol is not user friendly. In order to convert 0.5 mmol substrate, 15 mL TFA (a very nasty and strong acid) has to be used as the solvent. I am wondering if the authors can screen a more regular solvent instead of TFA. Once that solvent is found, I would be happy to recommend the acceptance of this paper in Nature Communication considering the novelty of this work itself.*

Response: We are grateful to the reviewer for the constructive comment, and agree with the reviewer's opinion that the use of 15 mL trifluoroacetic acid as a solvent for 0.5 mmol-scale reaction is the drawback of this method.

Actually, when we started this project, we screen a variety of regular solvents for this reaction. Unfortunately, these solvents didn't work well for this reaction. Electron-rich solvents, such as dioxane, tetrahydrofuran, ethylene glycol dimethyl ether didn't give any the desired product because of they were easier to be oxidized or decomposed than TFA. For electron-deficient solvent, such as acetonitrile and ester, which are regularly used solvents for photocatalysis, gave poor yields probably because of low efficiency of TiO_2 -based nanocatalyst. Electron-deficient halogen-containing solvents such as CHCl_3 , CCl_4 , 1,2-dichloroethane, didn't work for this reaction either likely due to homolytic cleavage of carbon-halogen bonds facilitated under UV irradiation. Water as a solvent didn't give any product because water underwent oxidative decomposition instead of oxidative

decarboxylation of trifluoroacetic acid. For aromatic solvents such as nitrobenzene and dichlorobenzene, the excess amount of solvents resulted in dominant trifluoromethylation on aromatic ring of solvent. In response to the comment, these results from solvent screening were illustrated in Table S1 of Supplementary Information.

The common characteristic of heterogeneous nanocatalysts is the relative low concentration of catalytic reactive centers compared with homogeneous catalysts. As a result, TiO₂-nanocatalyst for this trifluoromethylation reaction required sufficient light-absorbing surface to capture sufficient amount of light irradiation and therefore obtain satisfying reaction yields. The use of 15 mL trifluoroacetic acid as a solvent allowed the reaction system to obtain sufficient light-absorbing surface. We want to mention that the fact that TFA in this reaction is reusable by means of easy distillation for its purification and collection, may make it acceptable to use trifluoroacetic acid as a solvent.

In spite of the drawback of use of 15mL trifluoroacetic acid as a solvent, our current photodriven decarboxylative trifluoromethylation reaction may open up an avenue to accessing external oxidant-free C-H functionalization reactions, which is the reason why we submitted this work to *Nature Communications*. Because efficient molecular photocatalysis system is able to improve quantum efficiency of photocatalytic reactions and therefore use other solvents in place of trifluoroacetic acid and significantly reduce volume of solvents, the discovery and development of highly efficient molecular photocatalysts to achieve decarboxylative C-H trifluoromethylation reaction using trifluoroacetic acid as a starting material with a broad range of substrate scope under mild conditions is the goal of our next project. I hope that the reviewer concurs.

Response to the comments of reviewer #3

Comments #1. *Su and Li reported a practical and novel photo-driven C-H trifluoromethylation reaction, which applied trifluoroacetic acid as trifluoromethyl source. Rh-modified TiO₂ nanoparticle as catalysis made the transformation occurred in the absence of stoichiometric amounts of external oxidants. It was well demonstrated that a photo-driven catalysis process is a promising way to achieve external oxidant-free C-H functionalization reactions. However, there are still some problems should be answered before the manuscript was accepted on Nat. Commun.*

Response: We sincerely appreciate the reviewer for the very positive recommendation on the manuscript.

Comments #2. *Although no stoichiometric amounts of oxidant are necessary for the reaction, excess trifluoroacetic acid (15 mL) is needed. Please state the reason in manuscript (SI, Table S1, entry 2-17).*

Response: Thanks for a good suggestion. The common characteristic of heterogeneous nanocatalysts is the relative low concentration of catalytic reactive centers compared with homogeneous catalysts. As a result, the TiO₂-nanocatalyst used here for this trifluoromethylation reaction required sufficient light-absorbing surface to capture sufficient amount of light irradiation and therefore obtain satisfying reaction yields. The use of 15 mL trifluoroacetic acid as a solvent allowed the reaction system to obtain sufficient light-absorbing surface. In response to this comment, we added some sentences to the revised manuscript to state the reason for use of excess trifluoroacetic acid as a solvent. The added sentences are “In the established reaction, the use of 15 mL trifluoroacetic acid as a solvent may result from the need for sufficient light-absorbing surface to obtain satisfying yields presumably due to relative low concentration of catalytic active centers of heterogeneous TiO₂-based nanocatalyst”.

Comments #3. *The effects of substoichiometric amounts of oxidant should be explained in the mechanism (Figure 3, E) as a key advantage.*

Response: In the absence of external oxidant, this decarboxylative trifluoromethylation reaction gave the desired product, in which protons acted as oxidant for accepting the photo-

generated electrons. A substoichiometric amount of external oxidant (0.2 equivalents) played a similar role to protons in this reaction to accept the photo-generated electrons and drive reaction to proceed. Due to contribution of protons, a substoichiometric amount of external oxidant is sufficient to provide good yield for this reaction. In response to this comment, we have added the following sentence in the revised manuscript “Alternatively, the external oxidant introduced into the reaction system can also accept the photo-generated electron to drive the reaction. Consequent, combination of contributions of proton and a substoichiometric amount of external oxidant provided good yields for the photo-driven decarboxylative trifluoromethylation reaction”.

Comments #4. *The author declared that "the released protons accept the photo-generated electron to release hydrogen gas, a process similar to the photo-driven water splitting to H₂ evolution" (line 225 to 227). Here, the representative references should be included.*

Response: The representative reference has been cited in the revised manuscript in response to this comment.

REVIEWERS' COMMENTS:

Reviewer #1 (Remarks to the Author):

Authors have attended most of the queries from the reviewers satisfactorily.

Reviewer #2 (Remarks to the Author):

The manuscript has been revised carefully and it is now acceptable.

Reviewer #3 (Remarks to the Author):

In the revised version, Su and co-workers stated the reasonable and important reason for trifluoroacetic acid as solvent. The explanation, which "15 mL of TFA resulted from the need for sufficient light-absorbing surface to obtain satisfying yields presumably due to relative low concentration of catalytic active centers of heterogeneous TiO₂-based nanocatalyst", emphasized the concentration of nanocatalyst was significant for the light-absorbing. Meanwhile, the yield was low when other co-solvent is used in the same concentration (SI, Table SI). In my opinion, the reactive intermediate should be unstable in other solvents due to its high activity (E, Figure 3). TFA is the only effective solvent in this transformation.

All in all, the oxidative decarboxylative trifluoromethylation in absence of stoichiometric amounts of oxidant by means of photo-catalysis is a noteworthy advantage in this field. I recommend its publication in Nat. Commun.